# Transcriptomic Profiling of Circular RNA in Different Brain Regions of Parkinson’s Disease in a Mouse Model

**DOI:** 10.3390/ijms21083006

**Published:** 2020-04-24

**Authors:** Erteng Jia, Ying Zhou, Zhiyu Liu, Liujing Wang, Tinglan Ouyang, Min Pan, Yunfei Bai, Qinyu Ge

**Affiliations:** 1State Key Laboratory of Bioelectronics, School of Biological Science & Medical Engineering, Southeast University, Nanjing 210096, China; 230189589@seu.edu.cn (E.J.); 230198598@seu.edu.cn (Y.Z.); 230189168@seu.edu.cn (Z.L.); 220171806@seu.edu.cn (L.W.); 220171818@seu.edu.cn (T.O.); whitecf@seu.edu.cn (Y.B.); 2School of Medicine, Southeast University, Nanjing 210097, China; 101012091@seu.edu.cn

**Keywords:** Parkinson’s disease, circular RNAs, transcriptome, different brain region

## Abstract

Parkinson’s disease (PD) is the second most common neurodegenerative disease and although many studies have been done on this disease, the underlying mechanisms are still poorly understood and further studies are warranted. Therefore, this study identified circRNA expression profiles in the cerebral cortex (CC), hippocampus (HP), striatum (ST), and cerebellum (CB) regions of the 1-methyl-1,2,3,6-tetrahydropyridine (MPTP)-induced PD mouse model using RNA sequencing (RNA-seq), and differentially expressed circRNA were validated using reverse transcription quantitative real-time PCR (qRT-PCR). Gene ontology (GO), Kyoto Encyclopedia of Genes and Genomes (KEGG) pathway, and competing endogenous RNA (ceRNA) network analyses were also performed to explore the potential function of circRNAs. The results show that, compared with the control group, 24, 66, 71, and 121 differentially expressed circRNAs (DE-circRNAs) were found in the CC, HP, ST, and CB, respectively. PDST vs. PDCB, PDST vs. PDHP, and PDCB vs. PDHP groups have 578, 110, and 749 DE-circRNAs, respectively. Then, seven DE-cirRNAs were selected for qRT-PCR verification, where the expressions were consistent with the sequencing analysis. The GO and KEGG pathway analyses revealed that these DE-circRNAs participate in several biological functions and signaling pathways, including glutamic synapse, neuron to neuron synapse, cell morphogenesis involved in neuron differentiation, Parkinson’s disease, axon guidance, cGMP-PKG signaling pathway, and PI3K-Akt signaling pathway. Furthermore, the KEGG analysis of the target genes predicted by DE-circRNAs indicated that the target genes predicted by mmu_circRNA_0003292, mmu_circRNA_0001320, mmu_circRNA_0005976, and mmu_circRNA_0005388 were involved in the PD-related pathway. Overall, this is the first study on the expression profile of circRNAs in the different brain regions of PD mouse model. These results might facilitate our understanding of the potential roles of circRNAs in the pathogenesis of PD. Moreover, the results also indicate that the mmu_circRNA_0003292-miRNA-132-Nr4a2 pathway might be involved in the regulation of the molecular mechanism of Parkinson’s disease.

## 1. Introduction

Parkinson’s disease (PD) is a common progressive neurodegenerative disorder that affects millions of middle-aged and people older than 60 worldwide. Many factors affect the occurrence of PD, including genetic mutations, metabolism, diet, and environment [1]. The majority of PD cases are sporadic, while 5%–10% of PD is a monogenic form of PD, known as familial PD [2]. The main neuropathological hallmarks of PD include a loss of dopaminergic neurons in the substantia nigra and striatum [3]. The possible pathogenesis involves an increase in oxidative stress in the substantia nigra–striatum (ST) system; the excess free radicals under oxidative stress cause damage to dopamine (DA) [4]. In animal models, 1-methyl-1,2,3,6-tetrahydropyridine (MPTP) induces the selective destruction of dopaminergic neurons in the substantia nigra and striatum, and evokes symptoms of a Parkinsonian state, making it well-recognized in animal models of PD [5,6]. This change is associated with dyskinesia and early cognitive deficits [7]. During the 1960s, levodopa medications were the standard treatment for the motor symptoms of PD. These medications rapidly improve the motor symptoms, however, they cannot cure PD. Drug treatment cannot slow down or prevent the progress of neuron death in the dopamine system [8]. Unfortunately, there have not yet been other major breakthroughs in the treatment of PD, and levodopa is still the most commonly used drug to treat the disease. Therefore, to find efficient therapies for PD, researchers should study the pathogenesis of PD from the molecular biological level. In the context of the current aging global population, it is urgently necessary to study the molecular mechanism(s) underlying PD.

Circular RNA (circRNA) has become a research hotspot in recent years in the field of non-coding RNA. Compared with linear RNA, circRNA is more resistant to RNA exonuclease, and it is exceptionally stable in cells [9,10]. Consequently, circRNA may be more suitable for clinical marker research. CircRNA, long-chain noncoding RNA (lncRNA), and microRNA (miRNA) interact with each other, acting as competing endogenous RNAs (ceRNAs). It is helpful for us to explore the pathogenic genes and the mechanisms involved in the transcriptional regulatory network. It is known that ceRNAs participate in the transcriptional level and post-transcriptional modulation of disease-related target genes [11,12]. The functions of most circRNAs in different species remain unknown.

Recent studies have shown that many circRNAs play important roles in the regulation of gene expression at both transcriptional and post-transcriptional levels, and in the pathogenesis and progression of neurodegenerative diseases [13,14,15,16,17]. For example, circRNAs are abundantly expressed in the central nervous system, and play an important role in striatal synaptic plasticity and neuronal dysfunction [18,19,20]. Hansen et al. reported that circular transcript cerebellar-degeneration-related protein 1 antisense RNA (CDR1as) dysfunction resulted in increased levels of miR-7 targets, which in turn may affect brain function [21]. This indicates that circRNA participates in endogenous RNA to bind miRNA specifically, which thus is involved in the development of many diseases by regulating the expression of miRNA downstream target gene [22,23]. CircRNA is also involved in the pathogenesis of many neurological diseases, including Moyamoya disease [24], Alzheimer’s disease [25], periventricular white matter damage [26], and blood–brain–barrier integrity [27]. These studies have shown that circRNAs may be involved in the pathogenesis of neurological diseases, but more studies are needed to confirm this hypothesis. Based on the characteristics of stability, expression specificity, and participation in gene regulation, circRNAs are expected to be biomarkers for research on neurodegenerative diseases and aging.

Although mRNA, lncRNA, and circRNA are studied in PD [28,29], the mechanism of circRNAs in the pathogenesis of different brain regions has not been systematically studied. The function of the brain is not realized by a certain brain region or some connections, but by the interaction of different brain regions. Knowland et al. have shown that the pathogenesis of neurodegenerative diseases is closely related to impaired function in different brain regions [30]. Therefore, a comprehensive study of the functions of circRNAs in different brain regions is an advantageous strategy to understand neurodegenerative diseases such as PD. In this study, we performed RNA sequencing of varying brain regions to construct circRNAs expression profiles of the cerebral cortex (CC), hippocampus (HP), striatum (ST), and cerebellum (CB) of PD; screen differentially expressed circRNAs; and reveal their biological functions and molecular mechanisms, combined with bioinformatics analysis. The circRNAs’ expression and enrichment analysis of pathways would greatly facilitate the study of PD pathogenesis and provide potential novel targets for PD therapeutics.

## 2. Results

### 2.1. MPTP-Induced PD Motor Deficits

We evaluated the exercise capacity of MPTP-induced mice using both the pole and rotarod tests. Figure 1A,B shows the times taken for the mice to complete the pole test and rotarod test, respectively. In the pole test, the mice in the MPTP-induced model group took a longer time than in the control group (Figure 1A). In the rotarod test, compared with the control group mice, the MPTP-induced mice exhibited poor coordination, and their test time was significantly less (Figure 1B). This indicates that the MPTP-induced PD mouse model was successfully induced.

### 2.2. CircRNAs Expression Profiling in Different Brain Regions

To understand the circRNA information of different brain regions in PD mice, 24 libraries were constructed for the CC, HP, ST, and CB regions. The remaining RNA content after rRNA enrichment is shown in Appendix A. In this study, the sequencing metrics of each sample are listed, including the total reads, mapped reads, mRNA reads, rRNA ratio, circRNA reads, number of exonic reads, the number of paired reads, and the number of circRNAs detected (Appendix A). Among them, the total read range of the samples was 19 to 46 million, while the average circRNA reads in the samples were about 36,000. The number of circRNAs detected was about 5300 (Appendix A). In addition, we introduced a parameter—circular to linear ratio (CLR)—for the detected circRNAs to show their relative abundances [31] (Appendix A).

We perform hierarchical cluster analysis using an online analysis tools iDEP in each group of samples [32], and counts metric were used for this analyses; the data were normalized with FPKM (Figure 2A). The expression levels of circRNAs in different brain regions of the PD and control (CN) groups were compared, and the differentially expressed circRNAs (DE-circRNAs) were obtained on the basis of the following criteria: Log2 |fold change| ≥1, corrected *p*-values < 0.05. Then, an unbiased hierarchical clustering analysis was performed for the DE-circRNAs in the CC, HP, ST, and CB regions. The data for DE-circRNAs in the CC, HP, ST, and CB brain regions were visualized using a heat map (Figure 2B–E). The source and distribution of the DE-circRNAs are shown in Figure 2F. In the CC, HP, ST, and CB regions, most of the DE-circRNAs were distributed in the exon region (Figure 2G). The DE-circRNAs of the CC, HP, ST, and CB regions are shown in Appendix A, respectively. Figure 2H shows the overlap of DE-circRNAs in the four brain regions. In comparison with the healthy control, there were 24 DE-circRNAs in the CC (11 circRNAs up-regulated and 13 circRNAs down-regulated), 66 in the HP (25 circRNAs up-regulated and 41 circRNAs down-regulated), 71 in the ST (34 circRNAs up-regulated and 37 circRNAs down-regulated), and 121 in the CB (54 circRNAs up-regulated and 67 circRNAs down-regulated) (Figure 2I). There are unique circRNAs in different brain regions, including 22 in cerebral cortex, 54 in hippocampus, 60 in striatum and 113 in cerebellum. Recurrent circRNAs are rare in samples from different brain regions. We analyzed the unique circRNAs and recurrent circRNAs by GO and KEGG, and the results showed that the biological functions involved in unique circRNAs are consistent with our current main research results. However, recurrent circRNAs have not been enriched in biological functions and pathways.

### 2.3. Comparison of Expression Profiles in Different Brain Regions of PD Mice Model

The hierarchical cluster analysis revealed the circRNA expression levels in the PDST vs. PDCB, PDST vs. PDHP, and PDCB vs. PDHP groups (Figure 3A–C). The Venn diagram analysis showed the DE-circRNAs in the PDST vs. PDCB, PDST vs. PDHP, and PDCB vs. PDHP groups, among which 11 DE-circRNAs were overlapping (Figure 3D). The results showed that a total of 578 circRNAs were differentially expressed between the PDST and PDCB groups, with 240 being up-regulated and 338 being down-regulated (Figure 3E). At the same time, compared with the PDHP group, the PDCB group screened 749 DE-circRNAs, with 469 being up-regulated and 280 down-regulated (Figure 3E). A total of 110 DE-circRNAs were found to be significantly differentially expressed between the PDST and PDHP groups (Figure 3E). In the PDST vs. PDCB, PDST vs. PDHP, and PDCB vs. PDHP groups, the top 20 DE-circRNAs are listed in Appendix A, respectively.

### 2.4. Gene Ontology (GO) and Kyoto Encyclopedia of Genes and Genomes (KEGG) Analyses of the circRNAs Parental Genes in Different Brain Regions

To predict the function of circRNAs in the pathogenesis of PD, the DE-circRNAs were analyzed from three aspects, namely biological processes, cellular components, and molecular functions. The results indicate that 18 GO terms were significantly dysregulated (*p* < 0.05) in the ST region, among which the most genes and significantly enriched GO term was glutamatergic synapse (Figure 4A,B). In the CB region, the most genes and significantly enriched GO term was cell morphogenesis, involved in neuron differentiation (Figure 4C,D). In the HP region, the GO term with the most genes and the most significantly enriched term was synaptic membrane (Figure 4E,F). In the CC region, only two GO terms were enriched, including plasma membrane protein complex and endocytosis (Appendix A). The most significantly enriched GO term was the plasma membrane protein complex (Appendix A). The KEGG analysis revealed eight significant enrichment pathways, among which Alzheimer’s disease and the calcium signaling pathway were significantly enriched in the HP region; phosphatidylinositol signaling system, autophagy (animal), and MAPK signaling pathway were significantly enriched in the ST region; Parkinson’s disease, Axon guidance, and cGMP-PKG signaling pathway were significantly enriched in the CB region (Figure 4G). None of the pathways were significantly enriched in the CC region.

In addition, we performed GO and KEGG pathway analyses on DE-circRNAs in the PDST vs. PDCB, PDST vs. PDHP, and PDCB vs. PDHP groups. The top 20 results with the most significant *p*-values in the GO and KEGG pathway are shown in Figure 5. These results indicate that many GO terms from the biological process, cellular component, and molecular function domains were consistent in the comparison group (Appendix A). In the cellular component, the most significantly enriched GO terms were the synaptic membrane, axon, and glutamatergic synapse. In the biological process, the most significantly enriched GO terms were neuron projection morphogenesis, plasma membrane bounded cell projection morphogenesis, and synapse organization. In the molecular functions, the most significantly enriched GO terms were calmodulin binding, small GTPase binding, and protein-domain-specific binding. Moreover, the KEGG analyses revealed 36 significant enrichment pathways corresponding to the target genes, among which the most significant pathways were the glutamatergic synapse, axon guidance, cGMP-PKG signaling pathway, and calcium signaling pathway (Appendix A). At the same time, we analyzed co-expressed circRNAs in the three groups using GO analysis. The results show that the biological functions of these circRNAs were mainly related to postsynapse and cation transmembrane transport (Appendix A).

### 2.5. Validation by Quantitative Reverse Transcription PCR (qRT-PCR)

To validate the DE-circRNAs identified by RNA-Seq, we selected three co-expressing circRNAs and four circRNAs that may be related to PD for RT-PCR detection. The results showed that mmu_circRNA_0003292 and mmu_circRNA_0000870 had the lowest expression in the ST, the highest expression in the CB, and mmu_circRNA_0000517 had the highest expression in the ST. The expressions of mmu_circRNA_0004144, mmu_circRNA_0000468, and mmu_circRNA_0013321 were significantly down-regulated, while that of mmu_circRNA_0001320 was up-regulated in the PD group (Figure 5). The results showed similar changing trends between the qPCR verification and RNA sequencing results (Appendix A), which indicate the reliability of the sequencing analysis results.

### 2.6. CircRNA-Targeted miRNA-mRNA Network Prediction and Annotation

Firstly, we identified mmu-circRNA-0003292 and mmu-circRNA-0001320 as sponges for miRNA-132 and miRNA-124, respectively. The previous literature has shown that miRNA-132 and miRNA-124 are involved in Parkinson’s disease [33,34,35]. Therefore, we think that mmu-circRNA-0003292 and mmu-circRNA-0001320 may play a role in the pathogenesis of PD. In addition, the KEGG analysis showed that mmu-circRNA-0005976, mmu-circRNA-0003328, mmu-circRNA-0005388, and mmu-circRNA-0012384 were involved in Parkinson’s-related pathways [36,37,38]. Therefore, we chose these six circRNAs to predict the potential interactions among the identified circRNAs, miRNAs, and target genes’ mRNA. Based on the TargetScan and miRTarBase databases, we constructed a network of circRNA–miRNA–mRNA interactions, including 6 circRNAs, 13 miRNAs, and 112 mRNAs (Figure 6A). The network can visually show circRNA/miRNA/mRNA interaction. All of these target genes have been confirmed by biological experiments. A KEGG analysis was carried out to explore the putative functions of 112 target genes. The results show that the most significant pathway of the target genes predicted by mmu_circRNA_0001320, mmu_circRNA_0005976, mmu_circRNA_0005388, and mmu_circRNA_0003292 were the “neurotrophin signaling pathway”, “PI3K-Akt signaling pathway”, and “FoxO signaling pathway”, respectively (Figure 6B). Our research also shows that mmu_circRNA_0003292 regulated the Nr4a2 expression by miR-132. Previous studies have shown that the miR-132 expression changes significantly in the differentiation of dopaminergic neurons [33].

## 3. Discussion

PD is an age-related neurodegenerative disease that is characterized by a movement disorder from a loss of nigrostriatal dopamine neurons. At present, understanding of the underlying pathogenesis of PD is still not clear, which inevitably results in difficulties for targeted therapy. Although drugs can reduce the symptoms of PD, they do not prevent the disease from progressing. Accordingly, it is urgently necessary to investigate the underlying molecular mechanisms of PD. Recently, circRNAs have attracted huge attention because of their involvement in various biological processes [39], but the relationship between circRNAs and PD is still unclear. In this study, high-throughput transcriptome sequencing was used to evaluate the expression of circRNAs in different brain regions of PD, in order to further understand another layer of complexity of the PD brain transcriptome. As far as we know, this is the first comprehensive transcriptome analysis of circRNAs’ expression profiles in different PD brain regions.

We used the CIRI tool to identify circRNAs from RNA-seq data. Systematic filtering by this algorithm ensures a low false-positive rate without sacrificing the sensitivity of small circRNA and nonexonic circRNA detection [40]. Chen et al. only used the CIRI tool to predict circRNAs, and verified their existence by selecting eight circRNAs; the results indicated that the circRNA candidates in the RNA-seq datasets originated from the CIRI analysis were reliable [41]. Only using one tool for the circRNA prediction may cause a false-positive, which is a limitation of the study. The results of this study showed circRNAs to be expressed differently in the CC, HP, ST, and CB, and found many specific circRNAs in each region (Figure 2B), of which mmu_circRNA_0001320 was highly expressed in CB, while mmu_circRNA_0004144, mmu_circRNA_0000468, and mmu_circRNA_0013321 were not highly expressed in ST. Previous studies have shown that circRNAs are most abundantly expressed in the brain, especially in the CB [18,31,42], which is consistent with our research results. The GO and KEGG analyses revealed that DE-circRNAs are mainly involved in glutamic synapse, Parkinson’s disease, axon guidance, photohadylinositol signaling system, etc. These biological functions and pathways are involved in the pathogenesis of PD [43,44,45,46]. It is suggested that circRNAs may be involved in the pathogenesis of PD. Feng et al. only studied the ventral midbrains of PD mice, and found that circDLGAP4 had a neuroprotective effect, which can exert neuroprotective effects via modulating the miR-134-5p/CREB pathway both in humans and mice [47]. The functions of circRNAs have not been studied in other brain regions. Therefore, a comprehensive analysis of the circRNA expression profiles in the different brain regions is helpful to understand the pathogenesis of PD.

To investigate the potential functions of DE-circRNAs, we also constructed a circRNA–miRNA–mRNA interaction ceRNA network, which included six circRNAs, 13 miRNAs, and 112 mRNAs. Previous studies have established that circRNAs might function as microRNA sponges, inducing target mRNA silencing, thereby regulating gene expression at the post-transcriptional level [13,14,15,16,17]. Herein, through an in silico analysis, we predict that mmu-circRNA-0003292 is a sponge of miRNA-132. miR-132 is an important molecule regulating embryonic stem cell differentiation into dopamine neurons by directly targeting the Nr4a2 expression [33]. The over-expression of miRNA-132 reduced the differentiation of dopamine neurons [33], leading to damage to the brain systems involved in spatial learning and memory [48], resulting in an array of cognitive disorders in mice [49]. Lungu et al. also observed that miRNA-132 was up-regulated in a PD rat model [34], which is consistent with our results. Our data also show that Nr4a2 was a direct target of miRNA-132. NR4A2 is a transcription factor for midbrain dopamine neuron development and differentiation, which is not only essential in the development of mensencephalic dopaminergic neurons and the maintenance of their functions, but may also play a role in the pathogenesis of PD [50]. These results further suggested that the mmu-circRNA-0003292/miRNA-132 axis might affect the pathogenesis of PD by regulating the expression level of NR4A2. As the mechanisms of circRNAs have not been well elucidated, none of the potential circRNA–miRNA–mRNA pathways included in this study have been reported previously. The circRNA/miRNA/mRNA interaction and its role in PD need to be experimentally verified in future research.

In addition, we also found that mmu_circRNA_0001320 interacted with miRNA-124. It is well known that the over-expression of miRNA-124 can promote endogenous brain repair mechanisms, induce neuronal migration to the striatum, reduce the loss of DA neurons in the striatum, and deplete dopamine transmitters, thereby improving the motor symptoms of PD [35,51,52]. Therefore, we believe that the up-regulation of mmu_circRNA_0001320 could cause the down-regulation of miRNA-124, and thus participate in neurodegenerative diseases. It is very important to find out the interaction network of circRNA–miRNA–mRNA in order to explain the interactions between the RNA molecules and their functions, which will help to further understand the molecular mechanisms of PD. More importantly, a KEGG analysis of the target genes of six DE-circRNAs in the network indicated that the target genes of mmu_circRNA_0003292, mmu_circRNA_0001320, mmu_circRNA_0005976, and mmu_circRNA_0005388 are involved in the PD-related processes and pathways. Although the exact functions of most DE-circRNAs are still poorly understood, the bioinformatics analysis provided important evidence and clues for further study in PD. The results of this research not only provide an important basis for further exploring the pathogenesis mechanism of PD and the choice of treatment targets, but also a basis for a precision medicine for the disease.

Finally, the MPTP-induced PD mouse model study has several strengths and limitations. On the one hand, MPTP produced selectivity and reproducibility lesions of the nigrostriatal dopaminergic pathway after its systemic administration. On the other hand, age, gender, weight, and genetic background may have influenced both the reproducibility and the extent of the MPTP lesion in mice. While MPTP-lesioned monkeys remain the gold standard for the pre-clinical testing of new therapies for PD, most of the studies geared toward unraveling the mechanisms underlying PD have been performed primarily in mice. Although the MPTP mouse model is not perfect, we believe that it is suitable for the question being investigated and many previous studies on PD have also used MPTP-induced mouse models of PD [53,54,55].

## 4. Materials and Methods

### 4.1. Animals and Establishment of the PD Mouse Model

Eight SPF healthy male C57BL/6J mice (12-week-old, body weight 23–25 g) were purchased from Shanghai Southern Model Biotechnology Co., Ltd. The mice were housed at 22 °C under a 12 h light/dark cycle, with free access to food and water. The mice were randomly divided into a control group and a model group, with four mice in each group. After three days of acclimation, the mice in the model group were submitted to the pole test and rotarod test so as to collect baseline data. Then, the mice in the control group were intraperitoneally injected with normal saline. The mice in the model group were intraperitoneally injected with an MPTP (Sigma, St. Louis, MO, USA) solution every day, at a dose of 20 mg/kg for 10 days. After 10 days, we first conducted the pole test. First, we fixed a ball 2.5 cm in diameter to the top of a pole (1 cm diameter and 60 cm length). Then, we placed the mouse on the top of the pole and recorded the time taken to climb the pole. Secondly, we conducted the rotarod test. The rotarod experiment needed to keep the balance of the disease when moving continuously on the roller and used the average value for 20 consecutive measurements. Finally, three mice with obvious impairment in their sports ability and three mice that were healthy were screened out for use in the subsequent experiments. The study was reviewed and approved by the Ethics Committee of Zhongda Hospital Southeast University.

### 4.2. Sample Collection in Different Brain Regions

The animals (six in total; three in the control group and three in the PD model group) were anesthetized with tribromoethanol (500 mg/kg; Sigma, Saint Louis, USA), and were killed by cervical dislocation after the establishment of the PD mouse model. Brain samples were dissected to isolate the CC, HP, ST, and CB regions. Previous studies have primarily focused on the transcriptome analysis of the ST and substantia nigra brain regions [56], but few have studied the HP, CC, and CB regions. Emerging data suggest that there are interactions between the dopaminergic system and the HP, and these are involved in synaptic plasticity, adaptive memory, and motivated behavior. Therefore, it is necessary to study the pathological changes in the HP in order to further understand cognitive dysfunction in PD. In addition, Middleton et al. showed that the CB–thalamus–CC pathway affects motor and cognitive functions, and is widely connected with CC via specific pathways [57]. Ichinohe et al. also confirmed that there is a specific pathway between CB and ST in rats [58], that is, the cerebello–thalamo–motor cortico–striatal pathway, and the cerebello–thalamo–striatal pathway was found to affect the function of the ST. Therefore, it is necessary to study the differentially expressed genes in the CC, HP, ST, and CB brain regions, and the relationship between the functions of the four brain regions. The separated brains were removed and placed in a 2 mL enzyme-free Eppendorf tube. The brains were immediately frozen in liquid nitrogen and stored at −80 °C until subsequent analysis.

### 4.3. Total RNA Isolation and cDNA Library Construction

Total RNA was extracted from the CC, HP, ST, and CB regions by using Trizol Reagent (Invitrogen, Carlsbad, CA, USA). The quality and quantity of total RNAs were determined by an Agilent 2100 Bioanalyzer (Agilent Technologies, Palo Alto, CA, USA). The RNA samples will be submitted to the following library preparation steps when their RNA integrity numbers (RIN) were greater than 8.0. Firstly, the RNA was purified with 1.8× magnetic beads (VAHTATM RNA Clean Beads, N412) after it was treated with RNase-free DNaes I (Qiagen, Hilden, Germany) to remove the genomic DNA. Then, the rRNAs were removed by using the NEBNext rRNA Depletion Kit (NEB #E6310) according to user manual. The cDNA library was constructed using a NEBNext^®^ UltraTM II Directional RNA Library Prep Kit for Illumina^®^. The first step was to use reverse transcriptase and random primers to convert the interrupted RNA fragments into the first cDNA strand; the second cDNA strand was synthesized with DNA polymerase I and RNase H. The second step was the end repair process and adapter ligation step. Following this, the adapter was purified. The third step was the polymerase chain reaction (PCR) enrichment of the adaptor-ligated DNA. PCR products were purified using NEBNext Sample Purification Beads. The cDNA library concentration was measured using a Qubit^®^ 2.0 fluorometer; each sample was mixed to the same quality. Finally, the library was successfully sequenced on an Illumina HiSeq 2500 (Illumina Inc., San Diego, CA, USA) sequencer with a paired-end pattern and insert sizes of 300 bp.

### 4.4. CircRNA Transcriptome Analysis

After sequencing, according to Kim’s report [59], the reads obtained by sequencing were compared to the mouse reference genome using Tophat2 software. Then, CIRI (version 2.05) was used for circRNA detection and quantification [40]. We compared the expression of the circRNAs in the PD group and control group using the DESeq2 R package (1.16.1). Log^2^ |fold change| ≥1 and *p* < 0.05 were considered as differentially expressed circRNAs (DE-circRNAs).

### 4.5. Differentially Expressed circRNAs Enrichment Analysis

Gene ontology (GO) and Kyoto Encyclopedia of Genes and Genomes (KEGG) enrichment analyses of the DE-circRNAs were performed using the Metascape [60]. The GO terms and KEGG pathways of different brain regions were explored, with *p* < 0.05 indicating statistically significant enrichment. The functions of the DE-circRNAs and major pathways involved were determined by analyzing the significant enrichment of the GO terms and KEGG pathways.

### 4.6. Quantitative Reverse Transcription PCR (qRT-PCR) Analysis

In this study, we used qRT-PCR to perform circRNA verification on the same samples so as to compare the RNA-Seq and qRT-PCR results. Firstly, we designed the primers for the selected circRNAs. The obtained circRNA sequence was a linear sequence opened from backsplice. Then, a new sequence was formed by placing the 150–300 bp sequence of 3’ on the head of 5’. According to the principle of the PCR primer design, the circRNA primers were designed across the circularization site. The forward and reverse primers were on the left and right sides of the circularization site, respectively. We designed three pairs of primers for the circRNA junctions for each circRNA, and selected the most suitable pair of primers. The primers are listed in Table 1. The reaction program was set as follows: for 30 s at 95 °C, and 40 PCR cycles (95 °C, 5 s and 60 °C, 34 s (fluorescence collection)). GAPDH was used as an endogenous reference gene. The relative expressions of circRNAs were calculated using the 2^−ΔΔ*C*t^ method.

### 4.7. Construction of circRNA–miRNA–mRNA Interaction Network

First, the DE-circRNAs with a Log_2_ |fold change| ≥1 were selected. As circRNAs can function as endogenous miRNA sponges and can regulate the translation of targeted mRNAs, the expressions of circRNAs and mRNAs should be positively correlated, and the expression of miRNAs should be negatively correlated, with circRNAs and mRNAs. Then, TargetScan was used to predict miRNA, and mirTarBase was used to predict the target mRNAs, and the circRNA–microRNA–mRNA interaction network was constructed.

### 4.8. Statistical Analysis

The experimental data were analyzed with SPSS 22.0 (SPSS Inc., Chicago, IL, USA) software. The pole test and rotarod test data were analyzed using Student’s *t*-test and were presented as mean ± standard deviation (SD). Statistical significance was defined as *p* < 0.05

## 5. Conclusions

Our study is the first to provide circRNA expression profiling in different brain regions of PD mouse model. We identified 24, 66, 71, and 121 DE-circRNAs in CC, HP, ST, and CB, respectively. Using GO and KEGG analyses, several DE-circRNAs were found to be involved in the biological functions and pathways associated with PD. Among them, mmu_circRNA_0003292 functions, such as miRNA-132 sponges, may regulate the occurrence of PD via Nr4a2. These DE-circRNAs may provide new insights into the pathogenesis of PD, and yield novel biomarkers and therapeutic targets.

## Figures and Tables

**Figure 1 ijms-21-03006-f001:**
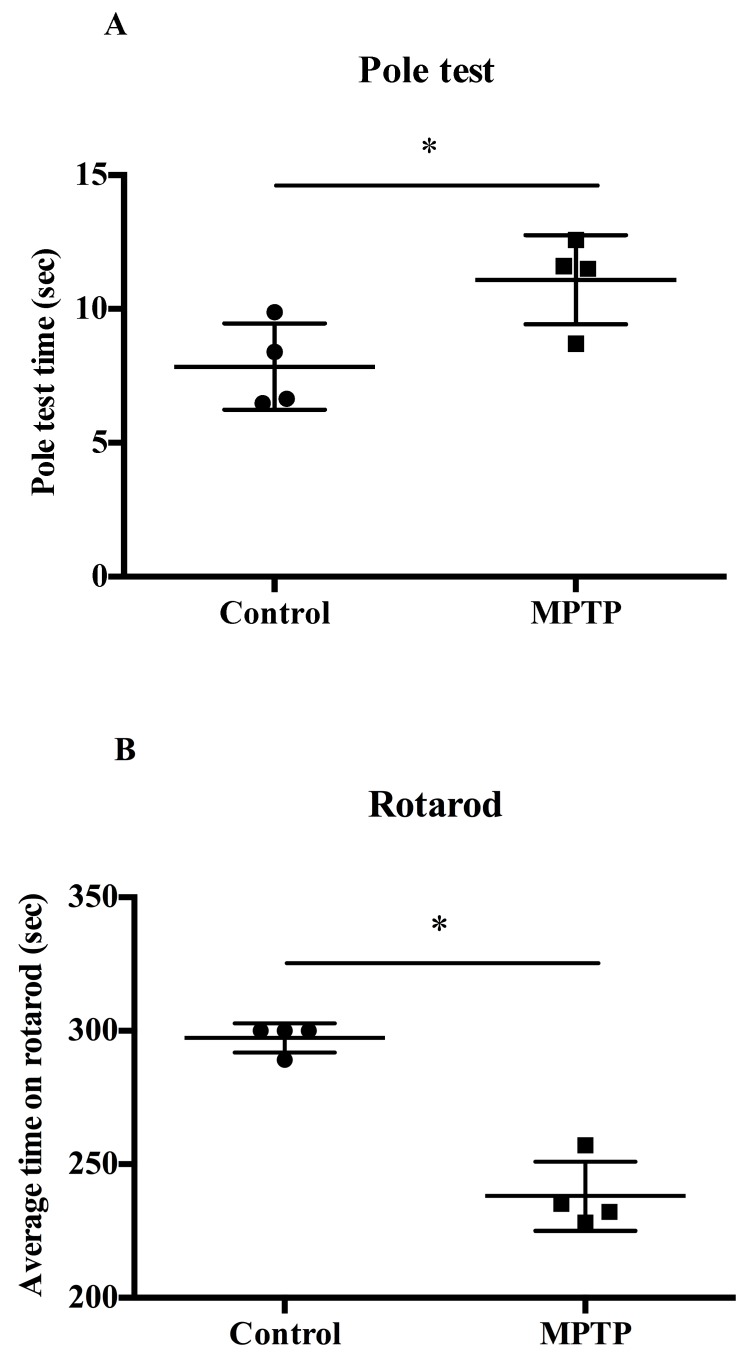
Measuring the exercise capacity of 1-methyl-1,2,3,6-tetrahydropyridine (MPTP)-induced model mice in a (**A**) pole test and (**B**) rotarod test. Values are expressed as mean ± standard error of the mean (SEM); * indicates a *p*-value < 0.05.

**Figure 2 ijms-21-03006-f002:**
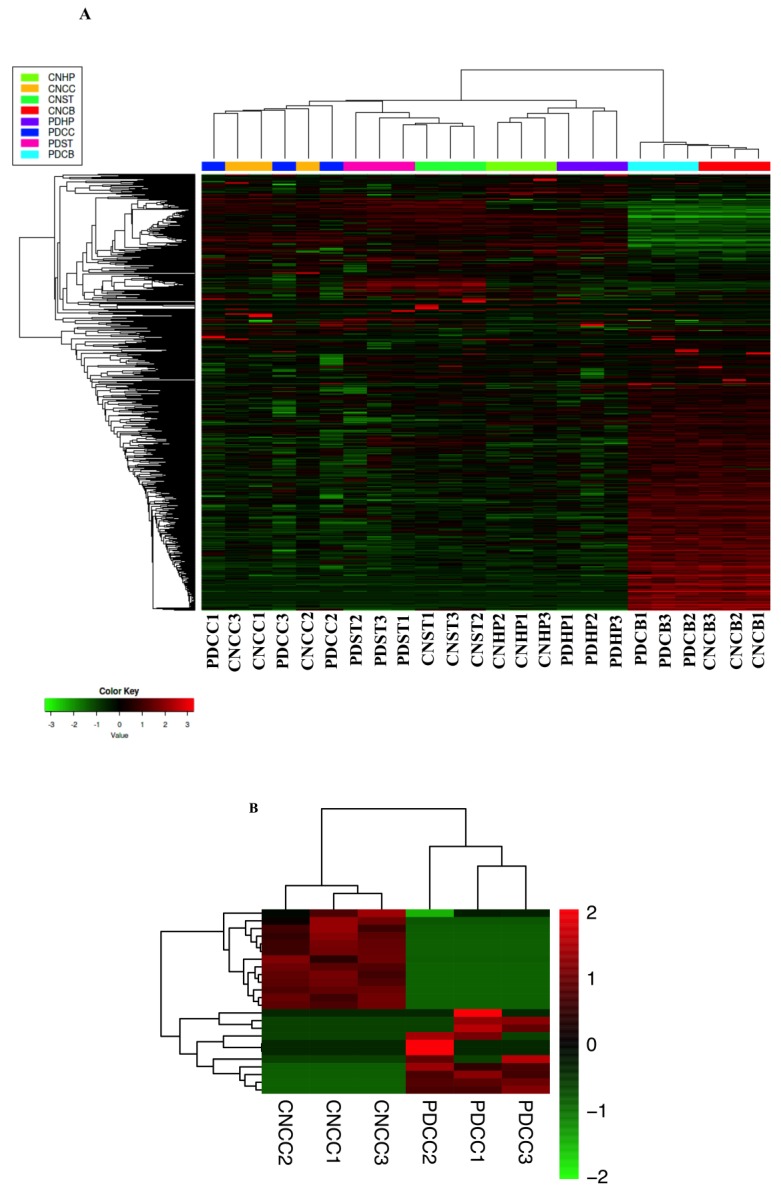
CircRNA expression profiling. (**A**) Heat map of circRNA expression in all samples in the different brain regions. (**B**) The heat map showing the expression of differentially expressed circRNAs (DE-circRNAs) in the cerebral cortex (CC). (**C**) The heat map showing the expression of DE-circRNAs in the hippocampus (HP). (**D**) The heat map showed the expression of DE-circRNAs in the striatum (ST). (**E**) The heat map showed the expression of DE-circRNAs in the cerebellum (CB). (**F**) Column graph of chromosome distributed DE-circRNAs to different groups according to the source of their host genes in the CC, HP, ST, and CB regions. (**G**) Based on the genomic origin, the histogram shows the classification of DE-circRNAs in the CC, HP, ST, and CB regions. (**H**) Overlap of significant DE-circRNAs in different brain regions. (**I**) DE-circRNAs analysis of different brain regions. CN stands for control samples.

**Figure 3 ijms-21-03006-f003:**
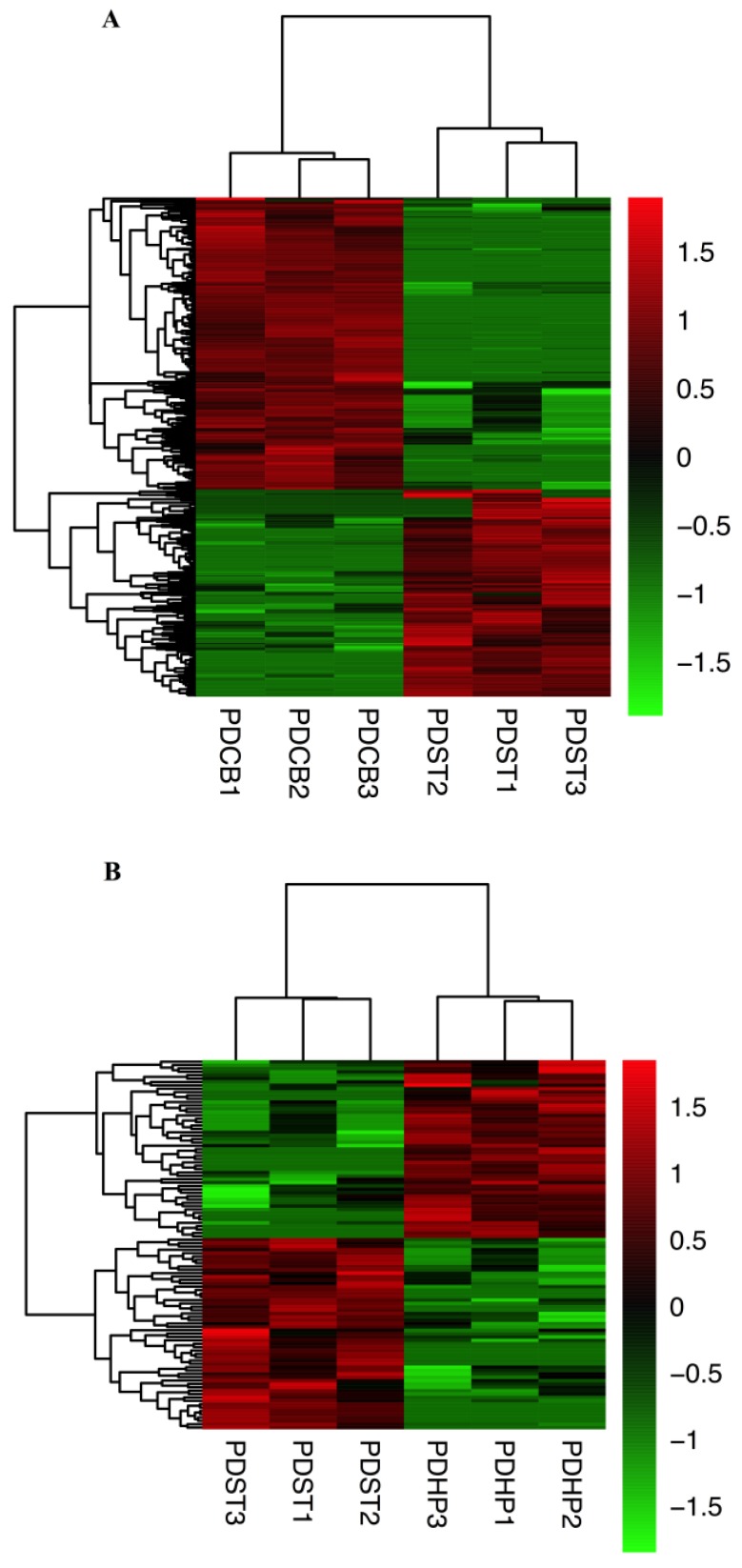
Differentially expressed circRNAs of PDST vs. PDCB, PDST vs. PDHP, and PDCB vs. PDHP groups. (**A**) The heat map of the DE-circRNAs for the comparison of PDST and PDCB. (**B**) The heat map of the d DE-circRNAs for the comparison of PDST and PDHP. (**C**) The heat map of the DE-circRNAs for the comparison of PDCB and PDHP. (**D**) Overlap of significant DE-circRNAs in PDST vs. PDCB, PDST vs. PDHP, and PDCB vs. PDHP groups. All values represent the number of DE-circRNAs. (**E**) DE-circRNAs analysis of PDST vs. PDCB, PDST vs. PDHP, and PDCB vs. PDHP groups.

**Figure 4 ijms-21-03006-f004:**
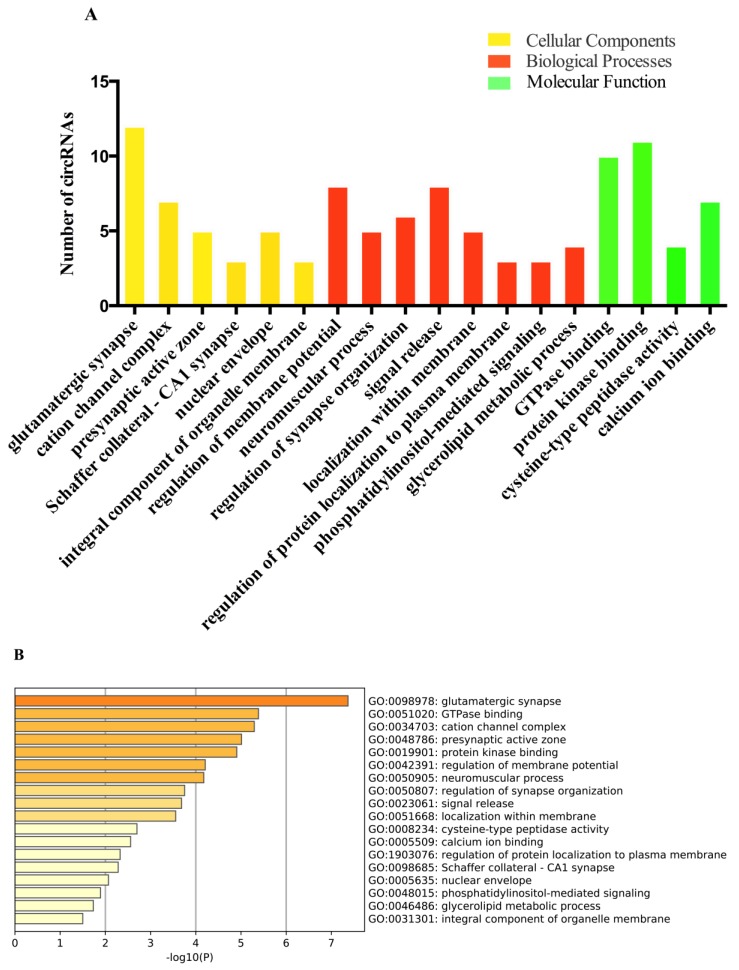
Gene ontology (GO) analyses of the DE-circRNAs parental genes in the ST and CB regions. (**A**) GO function classification of all DE-circRNAs in the ST region. (**B**) Bar chart of all DE-circRNAs clusters in the ST region. (**C**) GO function classification of all DE-circRNAs in the CB region. (**D**) Bar chart of the top 20 DE-circRNAs clusters in the CB region. (E) GO function classification of all DE-circRNAs in the HP region. (**F**) Bar chart of the all DE-circRNAs clusters in HP region. (**G**) The enriched KEGG pathways of the DE-circRNAs parental genes in different brain regions.

**Figure 5 ijms-21-03006-f005:**
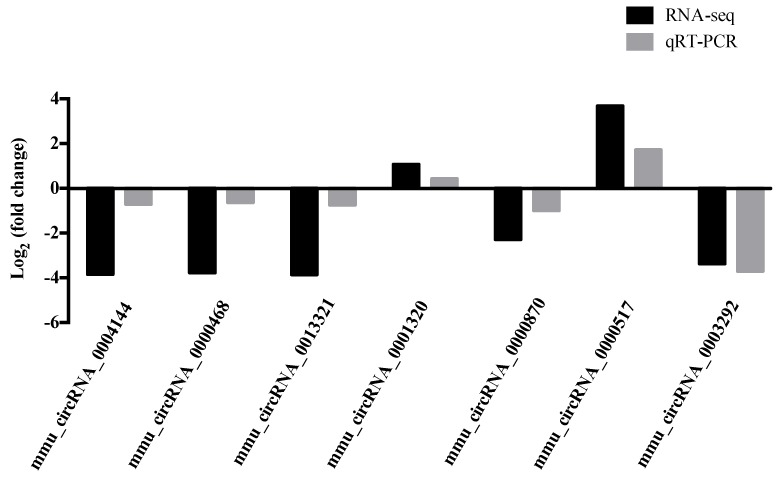
Quantitative real-time PCR validation of the selected circRNAs.

**Figure 6 ijms-21-03006-f006:**
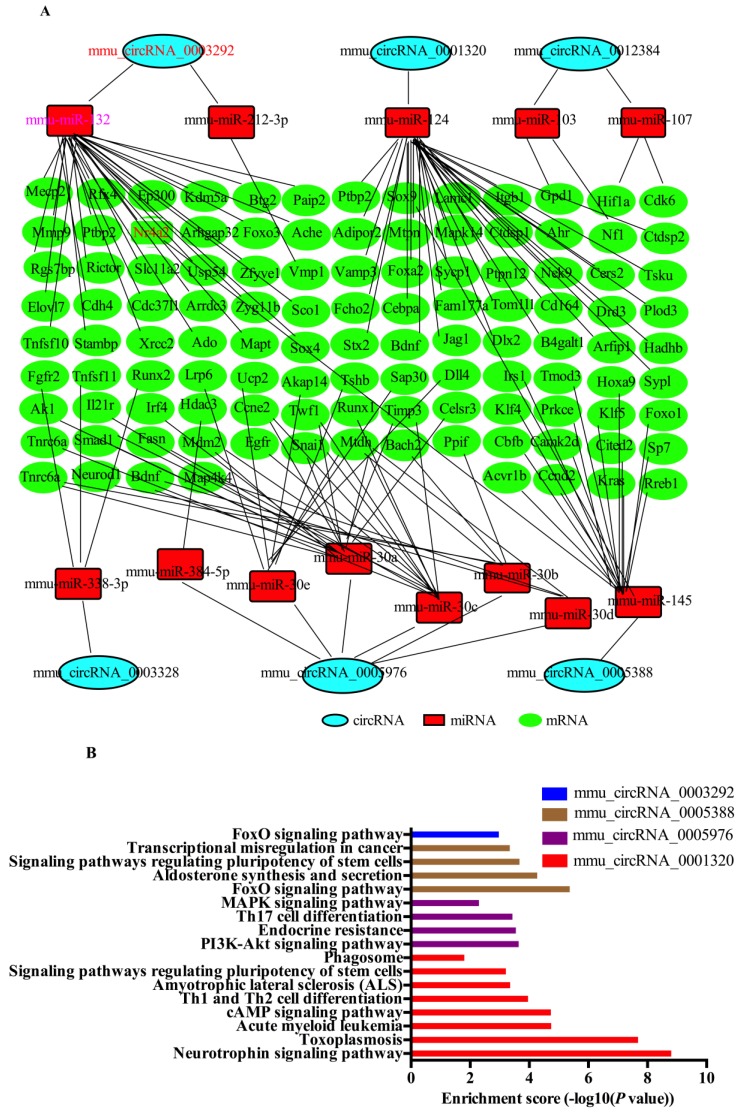
Prediction of DE-circRNAs-targeted genes based on TargetScan and miRTarBase. (**A**) The six predicted circRNAs-targeted circRNA–miRNA–mRNA networks. (**B**) The enriched Kyoto Encyclopedia of Genes and Genomes (KEGG) pathways of the DE-circRNAs targeted genes.

**Table 1 ijms-21-03006-t001:** The primers used in this study.

Gene	Sequences
GAPDH	F: 5’-CACTGAGCAAGAGAGGCCCTAT-3’ R: 5’-GCAGCGAACTTTATTGATGGTATT-3’
mmu_circRNA_0004144	F: 5’-TTAGCAGAGGAGCAAGCGTT-3’ R: 5’-TGCTCCTGAACCTGAAAATGT-3’
mmu_circRNA_0000468	F: 5’-CGTCACCAATCACACGGAGT-3’ R: 5’-CAGATGGCAGACCGTAGTCG-3’
mmu_circRNA_0013321	F: 5’-GTCTGACTGGTGGAACCCTG-3’ R: 5’-CCCAGAGGGATGGTGTAGCA-3’
mmu_circRNA_0001320	F: 5’-ACTCTGCTCGGGCGGT-3’ R: 5’-GCCGGCTTCGTGGATAATCT-3’
mmu_circRNA_0003292	F: GGTTTCACAGGTCTGGCGT R: CAACCGTTCCGTGGCTAACT
mmu_circRNA_0005157	F: GCAGCATTTGCAGGCACATAA R: AGCCTTGCCACACTCATCTC
mmu_circRNA_0000870	F: CGTCTAGAAGCATTGGGGCA R: CCACCATGAATAACTTTCACAAGC

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
