# Peer review of "Transcriptomic Profiling of Circular RNA in Different Brain Regions of Parkinson’s Disease in a Mouse Model"

_ijms, 2020, doi:10.3390/ijms21083006_

Round 1

Reviewer 1 Report

The authors describe circRNAs in four different brain regions of PD mice. They identify differentially expressed circRNAs in each brain region compared to a control group as well as compared to each other. In general, the paper is not very well written and presented, and lacks important aspects of circRNA analysis. My specific comments are below:

  1. There are several typographical and grammatical errors throughout the document, please ensure they are all corrected.
  2. Page 4, results section 2.2 - It is not clear what values were used to perform hierarchical clustering analysis? What kind of normalization, if any, was done for the number of circRNA supporting reads? Also, please indicate in the heatmap legends and in the text that CN stands for control samples.
  3. Page 8, results section 2.3 - The title of this section does not make sense, and it is not clear why CC samples were left out of these comparisons?
  4. The authors need to calculate the circular RNA to linear RNA ratios for the detected circRNAs to show their relative abundances. Common method for calculating this is described in Rybak Wolf et al., 2015 (Mol. Cell)
  5. Were there circRNAs that were uniquely detected in only one brain region and if so in how many samples were they recurrent in? Could we infer any biological significance for such unique and recurrent circRNAs?
  6. Page 14, section 2.5 - What was the criteria used to define “meaningful, enriched GO terms”?
  7. Page 17, section 2.6 - The criteria for selection of candidates for validation mentioned by the authors, i.e., candidates associated with PD - is not clear. How was their association with PD determined? In the discussion section they mention selecting circRNAs correlated with PD - how were they correlated (positive/negative) and how was this determined? Given that the authors detected so many DE-circRNAs, I would expect more number of candidates from high, medium and low expression ranges to be in the validation list.
  8. Page 20, section 2.7 - What was the criteria used to select the 6 circRNAs that were in the circRNA-miRNA-mRNA network? Target mRNAs were identified using TargetScan and mirTarBase but it’s not clear how the target miRNAs were identified?
  9. The discussion section repeats a lot of what is already described in the results section that is not necessary. In paragraph 2 of discussion section authors mention that DE circRNAs could be due to different distribution of circRNAs in the neuronal cells - but since no cell-specific analyses were performed, the authors need to make it clearer as to how they make this proposal?
  10. Page 24, section 4.1 - why were only 3 control mice used for the analyses? The authors only mention removing one mice from the experimental group. Also, how was the mice filtered out in the experimental group - can the authors clarify what “obvious impairment” means and how it is quantified/determined?
  11. Page 24, section 4.3 - what were the sequencing metrics? How many total reads, rRNA, mRNA reads were obtained? How much RNA was left after rRNA enrichment that went into library preparation?

Author Response

Thank you and the reviewer for giving us several critical suggestions to improve our manuscript. We have revised the manuscript thoroughly accordingly and addressed point by point below:

Responds to the reviewer1’s comments:

Comment 1: There are several typographical and grammatical errors throughout the document, please ensure they are all corrected.

Response: we have revised the whole manuscript carefully and tried to avoid any typographical and grammatical errors. Revised portion are marked in yellow in the revised manuscript.

Comment 2: Page 4, results section 2.2 - It is not clear what values were used to perform hierarchical clustering analysis? What kind of normalization, if any, was done for the number of circRNA supporting reads? Also, please indicate in the heatmap legends and in the text that CN stands for control samples.

Response: We perform hierarchical cluster analysis using an online analysis tools iDEP (http://bioinformatics.sdstate.edu/idep/) and counts metric were used for this analyses, the data were normalized with FPKM.

In addition, according to the reviewer’ suggestion, we have added in the heatmap legends and in the text that CN stands for control samples. Please see line 116 and 149 in the revised manuscript.

Comment 3: Page 8, results section 2.3 - The title of this section does not make sense, and it is not clear why CC samples were left out of these comparisons?

Response: Results section 2.3 is mainly to compare the correlation between the expression profiles of different brain regions in PD mice model. Now the title has been changed to “Comparison of expression profiles in different brain regions of PD mice model”. There are less differentially expressed circRNAs in the CC region; so, we focused on the analysis of three brain regions with more differentially expressed circRNAs for comparison with each other. Please see line 150 in the revised manuscript.

Comment 4: The authors need to calculate the circular RNA to linear RNA ratios for the detected circRNAs to show their relative abundances. Common method for calculating this is described in Rybak Wolf et al., 2015 (Mol. Cell)

Response: According to the reviewer’ suggestion, we have used the calculation method described by Rybak wolf et al. to calculate the circular RNA to linear RNA ratios (CLR) for the detected circRNAs in each sample. Please see Supplementary Data in the revised Supplementary Material.

Comment 5: Were there circRNAs that were uniquely detected in only one brain region and if so in how many samples were they recurrent in? Could we infer any biological significance for such unique and recurrent circRNAs?

Response: There are unique circRNAs in different brain regions, including 22 in cerebral cortex, 54 in hippocampus, 60 in striatum and 113 in cerebellum. Recurrent circRNAs are rare in samples from different brain regions. We analyzed the unique circRNAs and recurrent circRNAs by GO and KEGG, and the results showed that the biological functions involved in unique circRNAs are consistent with our current main research results. However, recurrent circRNAs have not been enriched in biological functions and pathways.

Comment 6: Page 14, section 2.5 - What was the criteria used to define “meaningful, enriched GO terms”?

Response: “meaningful, enriched GO terms” refers to the most significantly enriched GO terms, and we list the top 3 GO terms with the lowest p-values. GO analysis with false discovery rate (FDR)-adjusted p-values less than 0.05 were considered significantly enriched by DEGs. We have changed “meaningful, enriched GO terms” to “most significantly enriched GO terms”. Please see line 191 to 194 in the revised manuscript.

Comment 7: Page 17, section 2.6 - The criteria for selection of candidates for validation mentioned by the authors, i.e., candidates associated with PD - is not clear. How was their association with PD determined? In the discussion section they mention selecting circRNAs correlated with PD - how were they correlated (positive/negative) and how was this determined? Given that the authors detected so many DE-circRNAs, I would expect more number of candidates from high, medium and low expression ranges to be in the validation list.

Response: Among the 7 validated circRNAs, we randomly selected 3 co-expressing circRNAs and 4 circRNAs that may be related to PD. According to KEGG results, mmu_circ_0001320, mmu_circ_0004144, mmu_circ_0000468, and mmu_circ_0013321 target genes are involved in phosphatidylinositol signaling system. Previous studies have shown that phosphatidylinositol is an important substance for G-protein-coupled receptor signal transduction, and it can activate Ca+, Ca2+-dependent proteases and regulate various enzyme activities[1]. Ca2+/calmodulin-dependent protein kinase II (CaM-kinase II) is the most abundant protein kinase in mammalian brain neurons synapse, which is related to memory formation [2]. The variation of genes related to phosphatidylinositol metabolic pathway may lead to the development or function impairment of neurons, which is related to Parkinson's disease [3,4]. The target genes of these circRNAs play an important role in phosphatidylinositol signaling pathway, so we speculate that these circRNAs may be related to Parkinson’s disease.

However, in this study, the main purpose of PCR validation is to test the reliability of sequencing results.

[1] Valet, C.; Chicanne, G.; Severac, C.; et al. Essential role of class II PI3K-C2a in platelet membrane morphology. Blood 2015, 126(9):1128-1137.

[2] Armbrecht, H.J.; Siddiqui, A.M.; Green, M.; et al. SAMP8 mice have altered hippocampal gene expression in long term potentiation, phosphatidylinositol signaling, and endocytosis pathways. Neurobiol Aging 2014, 35(1):159-168.

[3] Duffy, C.; Kane, M.T. Investigation of the role of inositol and the phosphatidylinositol signal transduction system in mouse embryonic stem cells. Reproduction 1996, 108(1):87-93.

[4] Zubenko, G.S.; Hughes, H.B.; Jordan, R.M.; et al. Differential hippocampal gene expression and pathway analysis in an etiology-based mouse model of major depressive disorder. American Journal of Medical Genetics Part B Neuropsychiatric Genetics 2014, 165(6):457-466.

Comment 8: Page 20, section 2.7 - What was the criteria used to select the 6 circRNAs that were in the circRNA-miRNA-mRNA network? Target mRNAs were identified using TargetScan and mirTarBase but it’s not clear how the target miRNAs were identified?

Response: We chose circRNAs which may be related to Parkinson's disease to construct circRNA-miRNA-mRNA network.

Firstly, we identified mmu-circRNA-0003292 and mmu-circRNA-0001320 as a sponge for miRNA-132 and miRNA-124, respectively. The over-expression of miRNA-132 reduced dopamine neurons differentiation [1], leading to the damage of spatial learning and memory abilities [2], thereby leading to Parkinson's disease[3]. In addition, It is well known that over-expression of miRNA-124 can promote endogenous brain repair mechanisms, induce neuronal migration to the striatum, reduce the loss of DA neurons in the striatum and depletion of dopamine transmitters, thereby improving the motor symptoms of PD [4,5]. Therefore,we think that mmu-circRNA-0003292 and mmu-circRNA-0001320 may play a role in the pathogenesis of PD.

Secondly, ccording to KEGG analysis of differentially expressed circRNA-associated target genes, mmu_circRNA_0005976 and mmu_circRNA_0003328 are involved in the Parkinson's disease pathway, mmu_circRNA_0005388 is involved in the axon guidance pathway, and mmu_circRNA_0012384 target genes are involved in phosphatidylinositol signaling system. Axon guidance pathway is one of the critical processes related to connectivity and repair of the wiring of the brain during the central nervous system development and throughout the lifetime in humans[6]. Livesey et al. reported that genetic variability in the axon guidance pathway was a possible factor contributing to the development of one of the major neurodegenerative disorders, Parkinson’s disease (PD)[7]. The variation of genes related to phosphatidylinositol metabolic pathway may lead to the development or function impairment of neurons, which is related to Parkinson's disease[8,9]. In addition, Lrrk2 as a target gene of mmu_circRNA_0005976 has been shown to be related to Parkinson's disease [10,11]. Therefore, we speculated that mmu_circRNA_0005976, mmu_circRNA_0003328, mmu_circRNA_0005388, and mmu_circRNA_0012384 may play an important role in Parkinson's disease, so as to construct the circRNA-miRNA-mRNA network.

As for the target mRNA identified using TargetScan and mirTarBase, our description is wrong, TargetScan is used to predict miRNA, and mirTarBase is used to predict target mRNAs. We have modified this sentence. Please see line 225 and 373 to 375 in the revised manuscript.

[1] Yang, D.; Li, T.; Wang, Y.; Tang, Y.; Cui, H.; Tang, Y.; Zhang, X.; Chen, D.; Shen, N.; Le, W. miR-132 regulates the differentiation of dopamine neurons by directly targeting Nurr1 expression. J Cell Sci 2012, 125(Pt 7):1673-1682.

[2] Junn, E.; Mouradian, M.M. MicroRNAs in neurodegenerative diseases and their therapeutic potential. Pharmacol Ther 2012, 133(2):142-150.

[3] Lungu, G.; Stoica, G.; Ambrus, A. MicroRNA profiling and the role of microRNA-132 in neurodegeneration using a rat model. Neurosci Lett 2013, 553:153-158.

[4] Wang, H.; Ye, Y.; Zhu, Z.; Mo, L.; Lin, C.; Wang, Q.; Wang, H.; Gong, X.; He, X.; Lu, G.; Lu, F.; Zhang, S. MiR-124 Regulates Apoptosis and Autophagy Process in MPTP Model of Parkinson's Disease by Targeting to Bim. Brain Pathol 2016, 26(2):167-176.

[5] Saraiva, C.; Paiva, J.; Santos, T.; Ferreira, L.; Bernardino, L. MicroRNA-124 loaded nanoparticles enhance brain repair in Parkinson's disease. J Control Release 2016, 235:291-305.

[6] Tomiyama, H. A Commentary on Axon guidance pathway genes and Parkinson’s disease. journal of human genetics 2011, 56(2):102-3.

[7] Livesey, F.J.; Hunt, S.P. Netrin and netrin receptor expression in the embryonic mammalian nervous sys- tem suggests roles in retinal, striatal, nigral, and cere- bellar development. Mol. Cell Neurosci 1997, 8: 417–429.

[8] Valet, C.; Chicanne, G.; Severac, C.; et al. Essential role of class II PI3K-C2a in platelet membrane morphology. Blood, 2015, 126(9):1128-1137.

[9] Armbrecht, H.J.; Siddiqui, A.M.; Green, M.; et al. SAMP8 mice have altered hippocampal gene expression in long term potentiation, phosphatidylinositol signaling, and endocytosis pathways. Neurobiol Aging 2014, 35(1):159-168.

[10] Johannes, G.C.; Norbert, K.; Annette, S.; et al. The Parkinson disease causing LRRK2 mutation I2020T is associated with increased kinase activity. Human Molecular Genetics 2006, 15(2):223-232.

[11] Bjørg, J.W.; Aasly, J.O. Exploring cancer in LRRK2 mutation carriers and idiopathic Parkinson's disease. Brain and Behavior 2018, 8(5):e00858.

Comment 9: The discussion section repeats a lot of what is already described in the results section that is not necessary. In paragraph 2 of discussion section authors mention that DE circRNAs could be due to different distribution of circRNAs in the neuronal cells - but since no cell-specific analyses were performed, the authors need to make it clearer as to how they make this proposal?

Response: According to the reviewer’ suggestion, we have modified the description of some of the results of the discussion section. Please see line 242 to 265 in the revised manuscript.

   Previous literature has reported that there is a high abundance of circRNAs in the nerve tissue, and it shows tissue and cell specific distribution [1,2]. For example, during the development of pig brain, some circRNAs are highly expressed in the cerebellum, but not in the brainstem [3]. Therefore, we propose the hypothesis that DE-circRNAs could be due to different distribution of circRNAs in the neuronal cells. Maybe our conjecture is wrong.

 Now, according to the reviewer1/2’ suggestion, we have deleted conclusions about circRNA expression in neurons.

[1] Rybak-Wolf, A.; Stottmeister, C.; Gla?Ar, P.; et al. Circular RNAs in the Mammalian Brain Are Highly Abundant, Conserved, and Dynamically Expressed. Molecular Cell 2015, 58(5):870-885.

[2] Lyu, D.; Huang, S. The Emerging Role and Clinical Implication of Human Exonic Circular RNA. RNA Biology 2016, 14:1000-1006.

[3] You, X., Vlatkovic, I., Babic, A., Will, T., Epstein, I., Tushev, G., Akbalik, G., Wang, M., Glock, C., Quedenau, C., et al. Neural circular RNAs are derived from synaptic genes and regulated by development and plasticity. Nat Neurosci 2015, 18:603-610.

Comment 10: Page 24, section 4.1 - why were only 3 control mice used for the analyses? The authors only mention removing one mice from the experimental group. Also, how was the mice filtered out in the experimental group - can the authors clarify what “obvious impairment” means and how it is quantified/determined?

Response: We selected 3 mice with significantly reduced sports ability from 4 induced PD mice for subsequent experiments. In order to have the same number of repetitions as the PD model group, we only selected 3 mice in the control group. We have added information about the number of mice in the control group. Please see line 324 in the revised manuscript.

“obvious impairment” means significantly poor exercise capacity in their sports ability. We evaluated the exercise capacity of MPTP-induced mice using the pole test and rotarod test.

Comment 11: Page 24, section 4.3 - what were the sequencing metrics? How many total reads, rRNA, mRNA reads were obtained? How much RNA was left after rRNA enrichment that went into library preparation?

Response: We listed some metrics, including total reads, mapped reads, mRNA reads, rRNA ratio, circRNA reads, numer exonic reads, the number of paired reads, and the number of circRNAs detected. Please see Table S1 in the revised Supplementary Material.

Besides, we also analyzed the RNA content after rRNA enrichment, and the results are shown in Figure S1. Please see Figure S1 in the revised Supplementary Material.

Reviewer 2 Report

In this study, Jia et al. analyzed circRNAs expressed across 4 brain regions in six total mice (3 controls and 3 MPTP-treated mice). They identified differentially expressed circRNAs between the treated and control groups and across different brain regions. The reported results are derived from a very small sample set and the study would be strengthened if post-hoc meta-analyses were also performed using publicly available data. There are also numerous concerns I have summarized below, including the absence of RNase R depletion during sample processing and the absence of RNAseq analysis of predicted target mRNAs. Significant professional editing is also needed - it was difficult to follow the manuscript as there were many grammatical errors and the text was not always coherent.

Major comments:

1) Abstract: please describe what PD model was used

2) Intro: studies have shown that circRNAs may be involved with development of neurological diseases but more studies are needed to confirm this hypothesis.

3) How were the brain regions selected?

4) Methods: are the mice age-matched?

5) Methods: the authors describe that 3 of 4 mice demonstrated impairment (were these for only the MPTP-treated group?) – is this expected?

6) Methods: what was the quality of the extracted RNAs? Why was RNase R not used to deplete linear RNAs? This is currently the standard for assessment of circRNAs using RNAseq.

7) Methods: how many properly paired reads were sequenced for each sample?

8) Methods: given the known variability in circRNA detection across different tools, it is recommended to use an ensemble approach for circRNA detection. Other tools that can be used include find_circ, CIRCexplorer, etc.

9) Methods: for identification of differentially expressed circRNAs, did the authors try correcting P-values for multiple testing?

10) Methods: how was circRNA validation performed? Were the qRT-PCRs designed against circRNA junctions? Were replicates used?

11) For the supplementary tables, it would be helpful to include gene names associated with the Ensembl IDs, as well as the P-values, where applicable.

12) Section 2.4: how is it known if the identified circRNAs are associated with PD development or are associated with the endpoint of PD pathogenesis?

13) Figure 4: is there a reason why HP results are not shown here?

14) Figure 5 seems superfluous for the main manuscript. Suggest moving this to the supplementary section.

15) Sections 2.4 & 2.5: the text heavily describes what is already shown in Figures 4 & 5. Recommend making the Results discussion more concise.

16) Results: it would be helpful to include a supplementary table summarizing the validation results to show the log2 folds and P-values for each validated circRNA based on RNAseq data and based on qRT-PCR data to show correlation across data.

17) Section 2.7: how were the 6 circRNAs selected? Please provide references for the target genes that have been confirmed by biological experiments. Since the samples analyzed in the study were not depleted for linear RNAs, the mRNAs that are predicted to be regulated by circRNA expression ought to be assessed.

18) Given the findings shown in Figure 7, is there a reason why most of the circRNAs shown weren’t experimentally validated? It may be worth moving section 2.6 to after 2.7.

19) Discussion: since a cell-specific analysis was not performed, it is difficult to draw any conclusions about circRNA expression in neurons. Was circDLGAP4 identified in this study’s data set given the previously described study?

20) The authors state multiple times throughout the manuscript that they the analyses shed light into development of PD but this is speculative since only one time point was analyzed.

21) The Discussion does not read smoothly and jumps back and forth across different topics. Significant editing is needed.

22) Given the limited number of samples analyzed in this study, it would be worth querying existing public PD data sets to assess if the identified circRNAs (and/or minimally, the target mRNAs) are similarly observed in other models and samples.

Author Response

Thank you and the reviewer for giving us several critical suggestions to improve our manuscript. We have revised the manuscript thoroughly accordingly and addressed point by point below:

Responds to the reviewer2’s comments:

Comment 1: Abstract: please describe what PD model was used

Response: We have revised the manuscript according to the reviewer’ suggestion, the MPTP-induced PD mouse model used in this study. It could be seen in line 17 of revised manuscript.

Comment 2: Intro: studies have shown that circRNAs may be involved with development of neurological diseases but more studies are needed to confirm this hypothesis.

Response: We have revised the manuscript according to the reviewer’ suggestions. Please see line 77 to 78 in the revised manuscript.

Comment 3:  How were the brain regions selected?

Response: Previous studies have primarily focused on transcriptome analysis of ST and substantia nigra brain regions [1], but less have studied the HP, CC, and CB regions. Emerging data suggests that there are interactions between the dopaminergic system and the HP, and these are involved in synaptic plasticity, adaptive memory, and motivated behavior. Therefore, it is necessary to study the pathological changes in the HP in order to further understand cognitive dysfunction in PD. In addition, Middleton et al. showed that the CB–thalamus–CC pathway affects motor and cognitive functions and is widely connected with the CC via specific pathways [2]. Ichinohe et al. also confirmed that there is a specific pathway between the CB and ST in rats [3], that is, the cerebello–thalamo–motor cortico–striatal pathway, and the cerebello–thalamo–striatal pathway was found to affect the function of the ST. Therefore, it is necessary to study the differentially expressed genes in CC, HP, ST and CB brain regions and the relationship between the functions of the four brain regions.

[1] Alieva, A.K.; Zyrin, V.S.; Rudenok, M.M.; Kolacheva, A.A.; Shulskaya, M.V.; Ugryumov, M.V.; Slominsky, P.A.; Shadrina, M.I. Whole-Transcriptome Analysis of Mouse Models with MPTP-Induced Early Stages of Parkinson's Disease Reveals Stage-Specific Response of Transcriptome and a Possible Role of Myelin-Linked Genes in Neurodegeneration. Mol Neurobiol 2018, 55: 7229-7241.

[2] Middleton, F.A.; Strick, P.L. Basal ganglia and cerebellar loops: motor and cognitive circuits. Brain Res Brain Res Rev 2000, 31: 236-250.

[3] Ichinohe, N.; Mori, F.; Shoumura, K. A di-synaptic projection from the lateral cerebellar nucleus to the laterodorsal part of the striatum via the central lateral nucleus of the thalamus in the rat. Brain Res 2000, 880: 191-197.

Comment 4:  Methods: are the mice age-matched?

Response: Yes. We selected 12-week-old mice to induce mice model of PD. Previous studies have shown that many PD-related studies used 10-12-week-old mice to induce PD model [1-4]. So, the mice age are matched.

[1] Liu, Y.; Hao, S.; Yang, B.; et al. Wnt/β-catenin signaling plays an essential role in α7 nicotinic receptor-mediated neuroprotection of dopaminergic neurons in a mouse Parkinson’s disease model. Biochemical Pharmacology 2017, 140: 115-123.

[2] Zhang, Q.S.; Wang, Z.H.; Zhang, J.L.; et al. Beta-asarone protects against MPTP-induced Parkinson’s disease via regulating long non-coding RNA MALAT1 and inhibiting α-synuclein protein expression. Biomedicine & Pharmacotherapy 2016, 83: 153-159.

[3] Wang, Z.H.; Zhang, J.L.; Duan, Y.L.; et al. MicroRNA-214 participates in the neuroprotective effect of Resveratrol via inhibiting α-synuclein expression in MPTP-induced Parkinson’s disease mouse. Biomedicine & Pharmacotherapy 2015, 74: 252-256.

[4] Hsu, C.Y.; Hung, C.S.; et al. Ceftriaxone prevents and reverses behavioral and neuronal deficits in an MPTP-induced animal model of Parkinson’s disease dementia. Neuropharmacology 2015, 91: 43-56.

Comment 5:  Methods: the authors describe that 3 of 4 mice demonstrated impairment (were these for only the MPTP-treated group?) – is this expected?

Response: Yes, only the MPTP-treated group demonstrated impairment. This isn’t expected. Due to individual differences in immune system function in mice, the success rate of MTPT-induced PD mouse model is not 100%.

Comment 6: Methods: what was the quality of the extracted RNAs? Why was RNase R not used to deplete linear RNAs? This is currently the standard for assessment of circRNAs using RNAseq.

Response: RNA integrity is used to show the quality of the extracted RNA. The results showed that the RNA integrity number (RIN) of all samples was greater than 8.0.

On the one hand, due to the limited number of samples, we want to perform multiple tests, so, linear RNA is not removed. On the other hand, we want to obtain linear RNA and circular RNA at the same time, so we did not use RNaseR treatment. The removal of linear RNA will be considered in future functional verification.

Comment 7: Methods: how many properly paired reads were sequenced for each sample?

Response: According to the reviewer’ suggestion, we have added the number of paired reads that were sequenced for each sample. Please see Table S1 in the revised Supplementary Material.

Comment 8: Methods: given the known variability in circRNA detection across different tools, it is recommended to use an ensemble approach for circRNA detection. Other tools that can be used include find_circ, CIRCexplorer, etc.

Response: Thank you for your suggestion. We agree the reviewer's good advice, and we will use the tools recommended by the reviewer in future data analysis.

Comment 9: Methods: for identification of differentially expressed circRNAs, did the authors try correcting P-values for multiple testing?

Response: Yes, we have tried correcting P-values for multiple testing.

Comment 10: Methods: how was circRNA validation performed? Were the qRT-PCRs designed against circRNA junctions? Were replicates used?

Response: We performed the circRNA validation with qRT-PCR which designed against circRNA junctions and three replicates used.

Comment 11: For the supplementary tables, it would be helpful to include gene names associated with the Ensembl IDs, as well as the P-values, where applicable.

Response: Thank you for your valuable advice. According to the reviewer’ suggestion, we have added gene names and P-values in supplementary tables. Please see Table S2, Table S3, Table S4, Table S5, Table S6, Table S7 and Table S8 in the revised Supplementary Material.

Comment 12: Section 2.4: how is it known if the identified circRNAs are associated with PD development or are associated with the endpoint of PD pathogenesis?

Response: By GO and KEGG enrichment analysis, we detected the biological functions and pathways involved in circRNAs, and then speculated the possible correlation between circRNA and PD based on the relationship between PD and these biological functions and pathways reported in relevant literature.

Comment 13: Figure 4: is there a reason why HP results are not shown here?

Response: We revised the manuscript according to the reviewer’s suggestion and the HP results are shown in Figure 4. Please see line 205 to 206 in the revised manuscript.

Comment 14: Figure 5 seems superfluous for the main manuscript. Suggest moving this to the supplementary section.

Response: According to the reviewer’ suggestion, we have put Figure 5 in the supplementary material. Please see Figure S3 in the revised Supplementary Material.

Comment 15: Sections 2.4 & 2.5: the text heavily describes what is already shown in Figures 4 & 5. Recommend making the Results discussion more concise.

Response: We revised the manuscript according to the reviewer’s suggestion. Please see line 170 to 179 and 186 to 195 in the revised manuscript.

Comment 16: Results: it would be helpful to include a supplementary table summarizing the validation results to show the log2 folds and P-values for each validated circRNA based on RNAseq data and based on qRT-PCR data to show correlation across data.

Response: Thank you for your advice. According to the reviewer’ suggestion, we have added a table to the supplementary materials, including log 2(fold change) and p-value of each circRNAs measured by qPCR and RNA-seq, and added a figure to the supplementary materials, including relative expression level by qPCR and RNA-seq. Please see Table S9 and Figure S5 in the revised Supplementary Material.

Comment 17: Section 2.7: how were the 6 circRNAs selected? Please provide references for the target genes that have been confirmed by biological experiments. Since the samples analyzed in the study were not depleted for linear RNAs, the mRNAs that are predicted to be regulated by circRNA expression ought to be assessed.

Response: We chose circRNAs that may be related to Parkinson’s disease to construct circRNA-miRNA-mRNA network.

Firstly, we identified mmu-circRNA-0003292 and mmu-circRNA-0001320 as a sponge for miRNA-132 and miRNA-124, respectively. The over-expression of miRNA-132 reduced dopamine neurons differentiation [1], leading to the damage of spatial learning and memory abilities [2], thereby leading to Parkinson's disease [3]. In addition, It is well known that over-expression of miRNA-124 can promote endogenous brain repair mechanisms, induce neuronal migration to the striatum, reduce the loss of DA neurons in the striatum and depletion of dopamine transmitters, thereby improving the motor symptoms of PD [4,5]. Therefore,we think that mmu-circRNA-0003292 and mmu-circRNA-0001320 may play a role in the pathogenesis of PD.

Secondly, ccording to KEGG analysis of differentially expressed circRNA-associated target genes, mmu_circRNA_0005976 and mmu_circRNA_0003328 are involved in the Parkinson's disease pathway, mmu_circRNA_0005388 is involved in the axon guidance pathway, and mmu_circRNA_0012384 target genes are involved in phosphatidylinositol signaling system. Axon guidance pathway is one of the critical processes related to connectivity and repair of the wiring of the brain during the central nervous system development and throughout the lifetime in humans [6]. Livesey et al. reported that genetic variability in the axon guidance pathway was a possible factor contributing to the development of one of the major neurodegenerative disorders, Parkinson’s disease (PD)[7]. The variation of genes related to phosphatidylinositol metabolic pathway may lead to the development or function impairment of neurons, which is related to Parkinson's disease [8,9]. In addition, Lrrk2 as a target gene of mmu_circRNA_0005976 has been shown to be related to Parkinson's disease [10,11]. Therefore, we speculated that mmu_circRNA_0005976, mmu_circRNA_0003328, mmu_circRNA_0005388, and mmu_circRNA_0012384 may play an important role in PD, so as to construct the circRNA-miRNA-mRNA network.

Since the samples analyzed in the study were not depleted for linear RNAs, we calculate the circular RNA to linear RNA ratios (CLR) for the detected circRNAs in each sample. Please see Supplementary Data in the revised Supplementary Material.

[1] Yang, D.; Li, T.; Wang, Y.; Tang, Y.; Cui, H.; Tang, Y.; Zhang, X.; Chen, D.; Shen, N.; Le, W. miR-132 regulates the differentiation of dopamine neurons by directly targeting Nurr1 expression. J Cell Sci 2012, 125(Pt 7):1673-1682.

[2] Junn, E.; Mouradian, M.M. MicroRNAs in neurodegenerative diseases and their therapeutic potential. Pharmacol Ther 2012, 133(2):142-150.

[3] Lungu, G.; Stoica, G.; Ambrus, A. MicroRNA profiling and the role of microRNA-132 in neurodegeneration using a rat model. Neurosci Lett 2013, 553:153-158.

[4] Wang, H.; Ye, Y.; Zhu, Z.; Mo, L.; Lin, C.; Wang, Q.; Wang, H.; Gong, X.; He, X.; Lu, G.; Lu, F.; Zhang, S. MiR-124 Regulates Apoptosis and Autophagy Process in MPTP Model of Parkinson's Disease by Targeting to Bim. Brain Pathol 2016, 26(2):167-176.

[5] Saraiva, C.; Paiva, J.; Santos, T.; Ferreira, L.; Bernardino, L. MicroRNA-124 loaded nanoparticles enhance brain repair in Parkinson's disease. J Control Release 2016, 235:291-305.

[6] Tomiyama, H. A Commentary on Axon guidance pathway genes and Parkinson’s disease. journal of human genetics 2011, 56(2):102-3.

[7] Livesey, F.J.; Hunt, S.P. Netrin and netrin receptor expression in the embryonic mammalian nervous sys- tem suggests roles in retinal, striatal, nigral, and cere- bellar development. Mol. Cell Neurosci. 1997, 8: 417–429.

[8] Valet, C.; Chicanne, G.; Severac, C; et al. Essential role of class II PI3K-C2a in platelet membrane morphology. Blood 2015, 126(9):1128-1137.

[9] Armbrecht, H.J.; Siddiqui, A.M.; Green, M.; et al. SAMP8 mice have altered hippocampal gene expression in long term potentiation, phosphatidylinositol signaling, and endocytosis pathways. Neurobiol Aging 2014, 35(1):159-168.

[10] Johannes, G.C.; Norbert, K.; Annette, S.; et al. The Parkinson disease causing LRRK2 mutation I2020T is associated with increased kinase activity. Human Molecular Genetics 2006, 15(2):223-232.

[11] Bjørg, J.W.; Aasly, J.O. Exploring cancer in LRRK2 mutation carriers and idiopathic Parkinson's disease. Brain and Behavior 2018, 8(5):e00858.

Comment 18: Given the findings shown in Figure 7, is there a reason why most of the circRNAs shown weren’t experimentally validated? It may be worth moving section 2.6 to after 2.7.

Response: The circRNAs selected were validated in 2.6 section to demonstrate the reliability of RNA sequencing. This is the main purpose of circRNAs verification. Therefore, not all circRNAs in Figure 7 were verified.

Comment 19: Discussion: since a cell-specific analysis was not performed, it is difficult to draw any conclusions about circRNA expression in neurons. Was circDLGAP4 identified in this study’s data set given the previously described study?

Response: According to the reviewer1/2’ suggestion, we have deleted conclusions about circRNA expression in neurons. CircDLGAP4 was identified in this study’s data set.

Comment 20: The authors state multiple times throughout the manuscript that they the analyses shed light into development of PD but this is speculative since only one time point was analyzed.

Response: We have changed unsuitable description “the development of PD” to “the pathogenesis of PD”. Please see line 75, 78, and 260 in the revised manuscript.

Comment 21: The Discussion does not read smoothly and jumps back and forth across different topics. Significant editing is needed.

Response: Thank you for pointing this out for me. We have modified discussion section. The modified contents are highlighted by yellow. Please see discussion section in the revised manuscript.

Comment 22: Given the limited number of samples analyzed in this study, it would be worth querying existing public PD data sets to assess if the identified circRNAs (and/or minimally, the target mRNAs) are similarly observed in other models and samples.

Response: We have referred to the relevant literature and data and modified relevant parts of the article, and indeed some reported circRNAs are consistent with the results of this study [1,2].

[1] Tan Q. CircRNA expression profile of manganese-incuced Parkinsonism revealed by microarray and circRNA regulation mechanism. Doctoral dissertation 2019.

[2] Feng, Z.; Zhang, L.; Wang, S.; Hong, Q. Circular RNA circDLGAP4 exerts neuroprotective effects via modulating miR-134-5p/CREB pathway in Parkinson's disease. Biochem Biophys Res Commun 2019, pii: S0006-291X(19)32228-4.

Round 2

Reviewer 1 Report

Comment #2: Details about the hierarchical clustering tool and normalization should be added to the text.

Comment #5, #8: This needs to be clarified in the manuscript

Comment #7: Criteria for selection of candidates for validation needs to be clarified in the manuscript

Author Response

Thank you and the reviewer for giving us several critical suggestions to improve our manuscript. We have revised the manuscript thoroughly accordingly and addressed point by point below:

Responds to the reviewer1’s comments:

Comments and Suggestions for Authors

Comment #2: Details about the hierarchical clustering tool and normalization should be added to the text.

Response: According to the reviewer’ suggestion, we have added details about the hierarchical clustering tool and normalization. Please see line 118 to 120 in the revised manuscript.

Comment #5, #8: This needs to be clarified in the manuscript

Response: We have clarified in the manuscript. Please see line 134 to 139, and 232 to 239 in the revised manuscript.

Comment #7: Criteria for selection of candidates for validation needs to be clarified in the manuscript

Response: We have clarified in the manuscript. Please see line 221 to 223 in the revised manuscript.

Reviewer 2 Report

The authors have addressed some of my comments and concerns. Professional editing is needed as there are grammatical errors throughout the manuscript and the text is not always coherent and requires re-reading to understand what is being stated.

Comment#3: the explanation as to why the brain regions were selected, along with references, should be included in the manuscript Methods.

Comment#6: The RIN data should be included in the Methods. The text that was added to section 2.2 is confusing. Was rRNA enrichment performed? Do the authors mean depletion? Enriching for rRNA does not align with the study. This also impacts Figure S1, which needs a more detailed legend to describe what is being shown.

Comment#7: Please add commas to all numbers in tables to help with readability.

Comment#8: this concern was not addressed. Because an ensemble approach was not performed, the false positive rate is likely high. At a minimum, the caveat with only using one tool for circRNA prediction should be described in the Discussion, along with referencing previous reports of sensitivity and specificity associated with only using one tool.

Comment#9: after correcting the P-values, were there no differentially expressed circRNAs? If so, this should be stated in the Results.

Comment#10: details around qRT-PCR validation should be included in the methods, including how primer design was performed (e.g. targeting circRNA junctions, size of the target region, # replicates, etc).

Comment#16: Fig5 and FigS5 are redundant. It may make sense to include qRT-PCR data across all plots in Fig5 and remove FigS5. In the Methods, it should also be clarified if validations were performed on the same samples that RNAseq was performed for.

Comment#17: The reasons why the 6 circRNAs were selected should be summarized in the Results (& include references).

Additional comments based on revisions:

1) Discussion: the authors state that they identified mmu-circRNA-0003292 as a sponge for miR-132 but is this a novel finding? If so, it should be clarified that this is an in silico prediction and no laboratory experiments were performed so it is only a prediction.

2) Intro: the authors state that they performed deep RNAseq but based on the RNAseq metrics, this is not the case.

Author Response

Thank you and the reviewer for giving us several critical suggestions to improve our manuscript. We have revised the manuscript thoroughly accordingly and addressed point by point below:

Responds to the reviewer2’s comments:

Comments and Suggestions for Authors

Comment#: The authors have addressed some of my comments and concerns. Professional editing is needed as there are grammatical errors throughout the manuscript and the text is not always coherent and requires re-reading to understand what is being stated.

Response: Our article has undergone English language editing by MDPI. Please see English-Editing-Certificate-17947 and "English_Editing-manuscript".

Comment#3: the explanation as to why the brain regions were selected, along with references, should be included in the manuscript Methods.

Response: We have revised the manuscript according to the reviewer’ suggestion. Please see line 353 to 364 in the revised manuscript.

Comment#6: The RIN data should be included in the Methods. The text that was added to section 2.2 is confusing. Was rRNA enrichment performed? Do the authors mean depletion? Enriching for rRNA does not align with the study. This also impacts Figure S1, which needs a more detailed legend to describe what is being shown.

Response: We have added RIN data in the Methods, and modified Figure S1. Before the construction of cDNA library, rRNA was depleted. According to the reviewer1’ suggestion, we add some sequencing metrics in section 2.2. Please see line 370 to 371 in the revised manuscript, and Figure S1 in the revised Supplementary Material.

Comment#7: Please add commas to all numbers in tables to help with readability.

Response: According to the reviewer’ suggestion, we have added commas to all numbers in tables. Please see Tables in the revised Supplementary Material.

Comment#8: this concern was not addressed. Because an ensemble approach was not performed, the false positive rate is likely high. At a minimum, the caveat with only using one tool for circRNA prediction should be described in the Discussion, along with referencing previous reports of sensitivity and specificity associated with only using one tool.

Response: According to the reviewer’ suggestion, we have described using one tool to predict circRNAs and added references in the Discussion. Please see line 268 to 274 in the revised manuscript.

Comment#9: after correcting the P-values, were there no differentially expressed circRNAs? If so, this should be stated in the Results.

Response: Response: After correcting the P-values, the differentially expressed circRNAs can be screened, but they are less than the original ones. It may be the reason for the small number of sample or the correction method, and the method for correcting P-values for multiple testing may be too strict, resulting in fewer differentially expressed circRNAs screened finally. We have stated in the Results. Please see line 123 to 124 in the revised manuscript.

Comment#10: details around qRT-PCR validation should be included in the methods, including how primer design was performed (e.g. targeting circRNA junctions, size of the target region, # replicates, etc).

Response: According to the reviewer’ suggestion, we have added details around qRT-PCR validation. Please see line 399 to 406 in the revised manuscript.

Comment#16: Fig5 and FigS5 are redundant. It may make sense to include qRT-PCR data across all plots in Fig5 and remove FigS5. In the Methods, it should also be clarified if validations were performed on the same samples that RNAseq was performed for.

Response: We have deleted Fig. 5 and Fig. S5 and remade Figure 5. In additon, we have added the same sample information as RNAseq to the Methods. Please see line 399 to 400, Table S9 and Figure 5 in the revised manuscript.

Comment#17: The reasons why the 6 circRNAs were selected should be summarized in the Results (& include references).

Response: We have revised the manuscript according to the reviewer’ suggestions. Please see line 232 to 239 in the revised manuscript.

Additional comments based on revisions:

Comment#1: Discussion: the authors state that they identified mmu-circRNA-0003292 as a sponge for miR-132 but is this a novel finding? If so, it should be clarified that this is an in silico prediction and no laboratory experiments were performed so it is only a prediction.

Response: This is a novel finding. Now, we have modified this sentence. Please see line 292 to 293 in the revised manuscript.

Comment#2: Intro: the authors state that they performed deep RNAseq but based on the RNAseq metrics, this is not the case.

Response: We have revised this sentence. Please see line 90 to 92 in the revised manuscript.

Round 3

Reviewer 2 Report

The authors have addressed the majority of my concerns but some questions arise based on the most recent edits.

Comment #6: Based on this response and looking at lines 372-375, was poly-A selection performed? So far, research has found that circular RNAs are not poly-adenylated unless the authors are reporting an entirely new aspect to circular RNAs that has not been reported before. This requires clarification.

Comment #9: The text added to lines 123-124 is confusing since multiple testing corrections do not affect expression levels.

Comment #17: In lines 221-223, they say that the circRNAs were “randomly” selected which is not true since they describe their selection process.

Author Response

Thank you and the reviewer for giving us several critical suggestions to improve our manuscript. We have revised the manuscript thoroughly accordingly and addressed point by point below:

Responds to the reviewer2’s comments:

Comment #6: Based on this response and looking at lines 372-375, was poly-A selection performed? So far, research has found that circular RNAs are not poly-adenylated unless the authors are reporting an entirely new aspect to circular RNAs that has not been reported before. This requires clarification.

Response: We have revised the ambiguity description in the material and methods section. In this present study, total RNA were submitted for rRNA depletion with NEBNext rRNA Depletion Kit firstly, and then all the RNAs including mRNA, circRNAs and other non-coding RNAs were submitted for reverse transcription and the following library preparation steps. So RNAs with poly-A were not selected in this study. Please see line 370 to 375 in the revised manuscript.

Comment #9: The text added to lines 123-124 is confusing since multiple testing corrections do not affect expression levels.

Response: We have revised the misrepresentation in lines 120-123. It is impossible to be affected for the expression levels of circRNAs, while the number of DE-circRNAs obtained was reduced.

Comment #17: In lines 221-223, they say that the circRNAs were “randomly” selected which is not true since they describe their selection process.

Response: We revised the unsuitable description in lines 220-221.

This manuscript is a resubmission of an earlier submission. The following is a list of the peer review reports and author responses from that submission.

Round 1

Reviewer 1 Report

Comment #2: Details about the hierarchical clustering tool and normalization should be added to the text.

Comment #5, #8: This needs to be clarified in the manuscript

Comment #7: Criteria for selection of candidates for validation needs to be clarified in the manuscript

Reviewer 2 Report

This article focuses on the role of non-coding RNA in PD. It is definitely a very interesting topic, that still needs major investigations. Only a handful of publications are already available. The research carried out in this paper is technically sound and all the experiments presented are of high quality and reliable.

Because of this I found that it is pitiful that this manuscript was written in such a poorly manner. The paper has a terrible syntax and poor use of the English language. The article is really difficult to read, not only for the misuse of the language but also for the logic and the word arrangement along the paper. This is true from the very beginning. The abstract is too articulate, too many details are given and results are poorly interpreted. The results are better described in the discussion session, rather than in the results session and lack of a consistent interpretations. Conclusions are also foggy. I believe the whole article needs to be rewritten in a way that makes it easier to understand. I recommend the use of a professional, scientific writer.

Reviewer 3 Report

The manuscript concerns a very interesting field in molecular biology, related to the growing list of regulatory non-coding RNAs and their possible role in specific cell functions. In particular, the authors focused on circular RNAs (circRNA) expression profiles in Parkinson’s disease (PD) in different brain regions of PD mouse model. The advances in next-generation RNA sequencing techniques and bioinformatics tools allowed the discovery and the in-depth analysis of many transcripts, contributing to the understanding of gene expression regulation both in physiological and pathological conditions.

To date, several ncRNAs including some circRNA seem to be involved in neurodegenerative disorders, so the main purpose of the research by Jia et al. is valid and their results could contribute to providing new insights into the mechanisms of the disorder.

Unfortunately, I believe that the manuscript is not suitable for publication because it contains several major problems.

The whole article is hastily written, with too many misspelling and grammatical errors, it is very hard to read. I found repetitive sentences in the text and sometimes the bibliographical references are not inherent to the main topic.

Most importantly, the results are presented in a total unclear way, the figures are too small some of them also at very low resolution, unreadable on the printed version as well as on computer screen. Others have captions that lack important informations.

The discussion contains interesting but underdeveloped argumentation. For example, the analysis is conducted in the mouse model but very often this consideration is not remarked, nor is discussed the validity, strengths, and limitations of a PD mouse model for this kind of research.

Furthermore, the materials and methods section reports too little details of the several analyses and experiments conducted.

In this way, the reader must struggle to understand the authors' considerations and conclusions and many issues remain unclear.

I strongly recommend the authors to perform a careful revision of the manuscript, with particular attention to the writing style and manuscript organization, the presentation of the results, the quality and clarity of figures and tables. I believe these steps are mandatory to improve the scientific value of the results and are necessary for publication.